# Efficient Collocation of GNSS Radio Occultation Soundings with Passive Nadir Microwave Soundings

Alex Meredith[1], Stephen Leroy[2], Lucy Halperin[1], and Kerri Cahoy[1]

[1]Department of Aeronautics and Astronautics, Massachusetts Institute of Technology, Cambridge, MA 02139, USA
[2]Atmospheric and Environmental Research, Lexington, MA 02421, USA

**Correspondence:** Alex Meredith (ameredit@mit.edu)

**Abstract.** Radio Occultation (RO) using the Global Navigation Satellite Systems (GNSS) can be used to infer atmospheric profiles of microwave refractivity in the Earth's atmosphere. GNSS RO data are now assimilated into numerical weather prediction models and used for climate monitoring. New remote sensing applications are being considered that fuse GNSS RO soundings and passive nadir-scanned radiance soundings. Collocating RO soundings and nadir-scanned radiance soundings, however, is computationally expensive, especially as new commercial GNSS RO constellations greatly increase the number of global daily RO soundings. This paper develops a new and efficient technique, called the "rotation-collocation method", for collocating RO and nadir-scanned radiance soundings in which all soundings are rotated into the time-dependent reference frame in which the nadir sounder's scan pattern is stationary. Collocations with RO soundings are then found when the track of an RO sounding crosses the line corresponding to the nadir sounder's scan pattern. When applied to finding collocations between RO soundings from COSMIC-2, Metop-B-GRAS, and Metop-C-GRAS and the passive microwave soundings of ATMS on NOAA-20, Suomi-NPP, and AMSU-A on Metop-B and Metop-C for the month of January, 2021, the rotation-collocation method proves to be 99.0% accurate and is hundreds to thousands of times faster than traditional approaches to finding collocations.

## 1 Introduction

Measurements made using radio occultation (RO) by the Earth's atmosphere of the transmitters of the Global Navigation Satellite Systems (GNSS) are now routine and important contributors to numerical weather prediction and atmospheric reanalysis (Cardinali and Healy, 2014; Banos et al., 2019, and references therein). GNSS RO data fills in large holes in global coverage left by the international network of radiosondes, anchors atmospheric analyses by virtue of its near-absolute accuracy (Gelaro et al., 2017; Hersbach et al., 2020), and provides cloud-free information on atmospheric water vapor in the middle to lower troposphere (Kursinski and Gebhardt, 2014; Mascio et al., 2021). GNSS RO measurements are typically inverted to yield profiles of the index of refraction, a quantity with contributions from atmospheric density, temperature and water vapor (Kursinski et al., 2000).

Collocations of GNSS RO atmospheric soundings with the soundings of cross-track scanners in low Earth orbit are useful for several reasons. First, the contributions of water vapor and nitrogen/oxygen to the index of refraction cannot be separated based

on RO measurement alone; rather, separating their contributions requires the assistance of outside constraints. The commonly used outside constraint is the forecast of a numerical weather prediction system (e.g. Healy and Eyre, 2000), but specialized algorithms have been proposed that implement constraints based on collocated remote soundings by different techniques. One such algorithm considers column water vapor as inferred from microwave radiometers (Xie, 2006; Wang et al., 2017). Second, RO data have been used as a benchmark for investigating the accuracy of other remote sensing instruments by virtue of the near absolute accuracy of the measurements. RO soundings have been compared to collocated microwave soundings in order to validate the microwave soundings (Schröder et al., 2003; Ho et al., 2007; Iacovazzi et al., 2020) because microwave radiance standards are far less accurate than the timing standards which underly the GNSS signals used. GNSS RO validation reduces concerns about the use of satellite microwave data for climate trend studies. Inter-comparison of RO and spectral thermal infrared sounders for the sake of validating the calibration of the infrared sounders has also been investigated (Feltz et al., 2017; Yunck et al., 2009). These studies find that collocation between GNSS RO soundings and soundings of passive nadir cross-track scanners is necessary.

The computation of collocation between RO and passive nadir scanners is computationally expensive. Collocation is defined by tolerance windows in the spatial and temporal separation between a pair of RO and passive nadir soundings. The most direct approach to collocation is to consider a large batch of RO soundings and a large batch of passive nadir scanning data in time periods $\Delta T$ significantly longer than the temporal separation window $\Delta t$ and calculating the temporal and spatial separations between every potential pair of RO and nadir-scanner soundings to find pairs that meet the collocation criteria defined by the tolerance windows. Because of the large numbers of passive nadir soundings involved, the computation of collocations is extremely expensive. The expense can be reduced by decreasing the time window on passive nadir scanner soundings to be considered to $\Delta T = 2\Delta t$, but no further optimization is possible. We refer to collocation approaches similar to this as a *brute force* method. Publicly available tools for collocating satellite data generally use brute-force approaches which are not specific to the geometry of collocating GNSS RO and nadir-scanner soundings, and instead use parallelization and cloud computing to speed up collocation-finding (Chung et al., 2022; Smith et al., 2022; Wang et al., 2022).

Another method of collocation is motivated by the density and pattern of passive nadir soundings: they are so dense that they leave no gaps in their coverage during nominal operations, and their coverage pattern can be predicted precisely using an orbit propagator. If the reference frame for a collocation is one in which the scan pattern is just a stationary line of soundings, then a collocation is found when the location of an RO sounding in the scan-pattern reference frame crosses the nadir scanner's scan line. The advantage of such an algorithm is that the actual geolocations and times of the passive nadir soundings need not be considered at all; only the geolocations and times of the RO soundings do. Consequently, the determination of collocations should be greatly accelerated over a brute force method. The algorithm for collocation involving rotation into the reference frame of the nadir scan pattern we refer to as the *rotation-collocation* method.

The rotation-collocation method implemented in this paper identifies RO soundings which cross the nadir scanner's scan line and predicts the approximate time and location of the closest nadir-scanner footprint to these RO soundings, but does not extract the real nadir-scanner footprints collocated with these RO soundings. In order to fairly compare the rotation-collocation method to brute force methods, the brute force methods implemented in this paper also do not extract the nadir-

scanner footprints associated with collocated RO soundings, and instead leverage early termination once a collocation is found for faster collocation-finding.

The rotation-collocation method promises a great increase in efficiency over any brute force collocation method, but two complications must be addressed. Each is associated with a key assumption of the rotation-collocation method, and the errors that result must be quantified. The first assumption is that the scan of the passive nadir scanner is defined precisely as a line

in its own reference frame. In actuality, rather than a simple line, the scan pattern is a co-linear set of footprints of finite, non-zero sizes and distorted elliptical shapes, with greater distortion at the ends of the scan. The second assumption is that a simple orbit propagator and a range of scan angles of the passive nadir scanner is sufficient to determine the coverage of the scan footprints. These assumptions can be validated and the associated errors quantified by direct comparison of a set of collocations determined by the rotation-collocation method to a set of collocations determined by a brute force method, the

latter serving as a truth standard. Rates of false positives and false negatives can be estimated. Once these complications are addressed, then all that remains is to compute how great an acceleration in computation is gained by the rotation-collocation method over a standard brute force method.

This paper is organized as follows. Section 2 contains a description of the brute force method and a theoretical exposition of the rotation-collocation method. Both will be applied to candidate data sets in order to validate the rotation-collocation method and to determine the acceleration gained by the rotation-collocation method. Section 3 describes the data sets that will be used

in the study and defines the experimental setup. Section 4 contains an analysis of the experiments, including a quantification of the daily numbers of collocations of RO soundings with passive nadir microwave soundings. Section 5 presents the final conclusions.

## 2   Approach and theory of the collocation algorithms

This section describes the details of the brute force and rotation-collocation algorithms. Collocations are defined as RO soundings that are separated from a passive nadir sounding by at most $\Delta t$ in time and $\Delta d$ in distance. We consider the time corresponding to each RO sounding to be the start time of the RO measurement, and consider the position corresponding to each RO sounding to be the ray perigee (tangent) point projected onto Earth's surface. First, the details of two approaches to the brute-force algorithm are presented, and then two approaches to the rotation-collocation algorithm are presented.

### 2.1   The brute-force algorithm

The brute-force algorithm uses two checks: a spatial check and a time check. Because the brute-force algorithm makes no approximation, the brute-force method is a truth metric against which the accuracy of our rotation-collocation methods can be evaluated. This subsection describes two implementations of the brute-force algorithm: the first implementation considers all soundings of the nadir scanner over the course of a day when searching for collocations with RO soundings, and the second

implementation improves efficiency by sorting the nadir scanner soundings in time and windowing the soundings to within $\Delta t$ in time of the RO sounding before searching for spatial collocations.

### 2.1.1 Brute-force #1: All nadir scan soundings

The first brute-force approach compares every RO sounding to every nadir scanner sounding over a one-day period, performing a spatial check and a time check for every RO-nadir scanner sounding pair. A generic RO sounding has latitude $\theta_{RO}$, longitude $\lambda_{RO}$, and sounding time $t_{RO}$; and a generic nadir scanner sounding has latitude $\theta_{NS}$, longitude $\lambda_{NS}$, and sounding time $t_{NS}$. The spatial and temporal checks for collocation are:

$$\sqrt{(\lambda_{NS} - \lambda_{RO})^2 \cos^2 \theta_{RO} + (\theta_{NS} - \theta_{RO})^2} < \frac{\Delta d}{R_E} \tag{1a}$$

$$|t_{NS} - t_{RO}| < \Delta t \tag{1b}$$

in which $R_E$ is Earth's radius at the equator and longitudes and latitudes have units of radians. Note that equation 1a assumes small separations, $\delta d \ll R_E$. The temporal check is performed first, which permits a minor speed optimization using early termination: the logging practices of typical nadir scanner instruments generally associate a single time for a fixed number of footprints, thereby permitting a brute force method to greatly reduce the number of time checks.

### 2.1.2 Brute-force #2: Search-sort

This approach is similar to that of the brute-force method discussed in §2.1.1 but with narrowed windowing in time. The spatial check remains the same as the one given in Equation 1a; however, this approach avoids a time check by time-sorting the nadir scanner data. For each RO sounding, we search for the nadir scanner data indices corresponding to the window $[t_{RO} - \Delta t, t_{RO} + \Delta t]$. Then, we poll the nadir scanner data falling in this time window—which is guaranteed to pass the time check—and perform only the spatial check when searching for collocations.

With $n$ as the total number of nadir scanner soundings, and $r$ as the total number of RO soundings, brute-force method #1 has a time complexity of $O(rn)$ for the time check. Sorting the nadir scanner data has a time complexity of $O(n \log n)$, where $n$ is the total number of nadir scanner soundings, and searching the nadir scanner data has a time complexity of $O(\log n)$, so brute-force method #2 has a time complexity of $O(r \log n) + O(n \log n)$ for the time check, as this method performs one initial sort of the nadir scanner data and then one search of the nadir scanner data for each RO sounding.

In most cases, this method is faster than brute-force method #1, but when the number of nadir scanner soundings is very large (e.g. $\log n > r$, with $n$ the number of nadir scanner soundings and $r$ the number of RO soundings), the time required to sort the nadir scanner soundings can become long enough that brute-force method #2 takes longer than brute-force method #1. Furthermore, both brute-force methods avoid performing spatial checks for nadir scanner soundings outside of the time window, so the number of spatial checks performed by both methods is the same. The spatial check is much slower than the time check, so as the time window grows and the number of spatial checks required grows, the time taken by brute-force method #2 approaches the time taken by brute-force method #1.

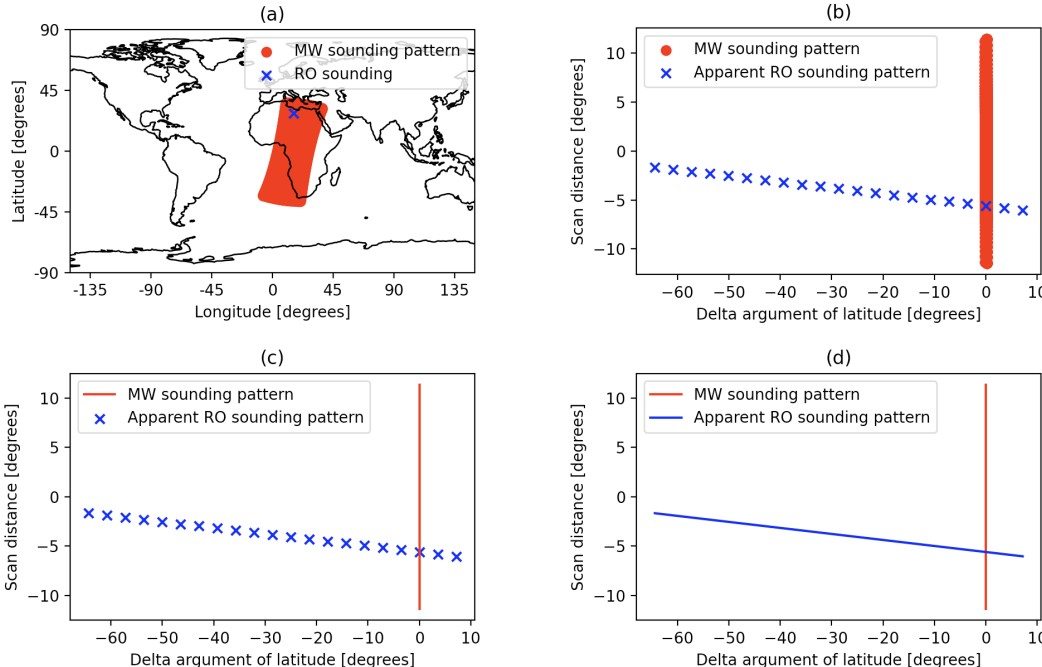

**Figure 1.** (a) A radio occultation from COSMIC-2 E1 plotted against contemporaneous MW soundings from NOAA-20 ATMS. (b) The discretized apparent position of the same occultation rotated into the MW frame, plotted against the MW sounding pattern. (c) The discretized apparent position of the RO sounding plotted against the linearized MW sounding pattern. (d) The interpolated apparent position of the RO sounding plotted against the linearized MW sounding pattern.

## 2.2 The rotation-collocation algorithm

The rotation-collocation method has two steps: first, rotating the RO soundings into the nadir scanning satellite's time-varying frame to find the apparent path of the RO sounding in the reference frame, and second, determining whether the apparent path of the RO sounding intersects the nadir scanner's pattern. The two patterns can only intersect if both the spatial and the temporal checks are satisfied.

Figure 1 illustrates the transformations undertaken by the rotation-collocation method. Panel (a) shows an RO sounding from COSMIC-2 E1 occurring at 00:23:52 UTC on January 2, 2021, and the pattern of NOAA-20 Advanced Technology Microwave Sounder (ATMS) soundings occurring within $\Delta t = 600$ seconds of the RO sounding, from 00:13:52 to 00:33:52 UTC. Panel (b) shows the COSMIC-2 RO sounding and the ATMS soundings rotated into the time-varying frame of NOAA-20. All of the ATMS soundings collapse to a near-perfect single line that extends upward and downward by an amount related to the range of the nadir scanner's scan angles. Also notice that the single RO sounding is represented as a series of points corresponding to its apparent location in the rotated frame at varying times $t_i$ with the time window $[t_{\mathrm{RO}} - \Delta t, t_{\mathrm{RO}} + \Delta t]$. In this work, we

refer to these apparent locations of a single RO sounding in the time-varying rotated frame as "sub-occultations", as shown in panels (b) and (c) of Figure 1. The apparent path of the sub-occultations crosses the line of the ATMS scan pattern, indicating the existence of a collocation.

Panel (c) of Figure 1 demonstrates the first approximation associated with the rotation-collocation algorithm, that a nadir scanner sounding pattern can be approximated by a perfect line at $\delta u = 0$ with $u$ the argument of latitude, or along-track coordinate of the nadir scanner, found using an orbit propagator. This approximation rests on three major assumptions: first, that the footprints of the nadir scanner, which are distorted ellipses, can be treated as single points; second, that the orbit propagator used in the rotation is perfectly accurate; and third, that the nadir scanner sounding pattern leaves no gap in coverage. This approximation has the advantage of not having to consider any of the geolocations of the nadir scanning instrument at all.

Panel (d) of Figure 1 illustrates the second approximation of the rotation-collocation algorithm, that the sub-occultations fall on a straight line in the rotated frame. Without this second approximation, the location of each sub-occultation must be computed and the line connecting consecutive sub-occultations must be checked for crossing the scan line of the nadir scanner. With the second approximation, however, only the sub-occultations at times $t_{\mathrm{RO}} - \Delta t$ and $t_{\mathrm{RO}} + \Delta t$ are computed and the line connecting the two checked for crossing the nadir scanner scan line. The only imperfection of this approximation is that there is some minute amount of curvature associated with the path of the RO sub-occultations in the rotated frame, and that curvature becomes increasingly pronounced with longer time collocation windows $\Delta t$.

The explicit rotation of the rotation-collocation algorithm is given by

$$\begin{pmatrix} x_R \\ y_R \\ z_R \end{pmatrix} = \begin{bmatrix} \cos u(t) & \sin u(t) & 0 \\ -\sin u(t) & \cos u(t) & 0 \\ 0 & 0 & 1 \end{bmatrix} \begin{bmatrix} 1 & 0 & 0 \\ 0 & \cos i & \sin i \\ 0 & -\sin i & \cos i \end{bmatrix} \begin{bmatrix} \cos \Omega(t) & \sin \Omega(t) & 0 \\ -\sin \Omega(t) & \cos \Omega(t) & 0 \\ 0 & 0 & 1 \end{bmatrix} \begin{pmatrix} x_{\mathrm{ECI}}(t) \\ y_{\mathrm{ECI}}(t) \\ z_{\mathrm{ECI}}(t) \end{pmatrix} \tag{2a}$$

in which $u$, $i$, and $\Omega$ are the argument of latitude, the inclination, and the right ascension of the ascending node of the nadir scanner satellite, and the coordinates $x_{\mathrm{ECI}}(t), y_{\mathrm{ECI}}(t), z_{\mathrm{ECI}}(t)$ are Cartesian coordinates of a location in an Earth-centered inertial (ECI) coordinate system. In the collocation problem, the input coordinates are longitude $\lambda$ and latitude $\theta$, and so first we transform the latitude and longitude of a sounding to a position in an Earth-centered Earth-fixed (ECF) coordinate system given by $(\cos \lambda \cos \theta, \sin \lambda \cos \theta, \sin \theta)$ and then compute the ECI coordinates according to the time-dependent transformation $\mathcal{L}_t$:

$$\big(x_{\mathrm{ECI}}(t), y_{\mathrm{ECI}}(t), z_{\mathrm{ECI}}(t)\big) = \mathcal{L}_t\big(x_{\mathrm{ECF}}, y_{\mathrm{ECF}}, z_{\mathrm{ECF}}\big) = \mathcal{L}_t\big(\cos \lambda \cos \phi, \sin \lambda \cos \phi, \sin \phi\big) \tag{2b}$$

The results of equations 2a and 2b are the rotated Cartesian coordinates $(x_R, y_R, z_R)$. These coordinates are best interpreted as an along-track coordinate that we call "delta argument of latitude" ($\delta u$) and the cross-track coordinate that we call the "scan distance" ($\delta s$):

$$\delta u = \arctan(y_R, x_R) \tag{3a}$$

$$\delta s = \arcsin z_R \tag{3b}$$

in which $\arctan(\cdots, \cdots)$ is a four-quadrant arctangent defined such that $\tan \delta u = y_R / x_R$. Both $\delta u$ and $\delta s$ are distances on the Earth's surface in units of radians. They can be converted to degrees by multiplying by $180°/\pi$ as in figure 1 or to distance by multiplying by the radius of the Earth ($R_E$).

The scan pattern of the nadir-scanning satellite in the rotated frame of reference can be described as the line segment $-\delta s_{\max} < \delta s < \delta s_{\max}$, $\delta u = 0$, where $\delta s_{\max}$ is essentially limited by $\xi_{\max}$, the maximum scan angle of the nadir-scanning instrument. The relationship between the maximum of the scan distance ($\delta s_{\max}$) in the rotated frame and the maximum scan angle ($\xi_{\max}$) of the scanning instrument is found using the law of sines:

$$\delta s_{\max} = \arcsin\left(\frac{a(t)}{R_E(\theta(t))} \sin \xi_{\max}\right) - \xi_{\max} \tag{4}$$

in which $a(t)$ is the radius of the nadir-scanner satellite's orbit at the time of the collocation. The radius $a(t)$ can be determined by finding the time $t$ at which the line connecting sub-occultations crosses the scan line of the nadir scanner, and then using *sgp4* (Vallado et al., 2006; Vallado and Crawford, 2008) to propagate the nadir-scanner orbit until time $t$.

The computation of the scan distance allows for a minor correction associated with the oblateness of the Earth, namely that the Earth's radius is a function of latitude, and nadir-scanner latitude is a function of time ($R_E = R_E(\theta(t))$). Including $a(t)$ in the computation, rather than using a constant orbital radius, allows for an additional minor correction for nadir-scanning satellites with nonzero eccentricity. Because the exact collocation time $t$ is initially unknown, the rotation-collocation algorithm initially calculates $R_E(\theta(t))$ and $a(t)$ using the occultation time, and then if a collocation is found, recalculates $R_E(\theta(t))$, $a(t)$, and $\delta s_{max}$ using the collocation time $t$ and performs a second follow-up check with the new, more precise value of $\delta s_{\max}$.

### 2.2.1   Rotation-collocation #1: Sub-occultations

In order to determine collocation, then, it is only necessary to check whether the path of the RO sounding in the rotated frame crosses the line associated with the scan pattern of the nadir-scanning instrument. Recall that the RO sounding is a trajectory in this frame because the coordinate system rotates with the scan line of the nadir scanner, which itself is moving during the time window $[t_{\mathrm{RO}} - \Delta t, t_{\mathrm{RO}} + \Delta t]$. We define the apparent trajectory of sub-occultations for a generic RO sounding with longitude $\lambda_{RO}$, latitude $\theta_{RO}$, and time $t_{RO}$ at times $t_i$ by

$$t_i = t_{\mathrm{RO}} + dt \left(i - (N+1)/2\right) \tag{5}$$

in which $dt = 2\Delta t/(N-1)$ is the time separation between consecutive sub-occultations and $N$ is the number of sub-occultations. The position of each sub-occultation is computed in the rotated frame (recall that the transformations of equations 2a and 2b are both time-dependent). Each segment connecting consecutive sub-occultations in the rotated frame is checked for crossing the scan line $\delta u = 0$ of the nadir-scanning instrument. If any segment crosses the scan line, the temporal check for collocation is satisfied. If the intersection occurs at a scan distance $|\delta s| < \delta s_{\max}$, then the spatial check for collocation is satisfied. When both the spatial and temporal checks are satisfied, a collocation is found.

The computational expense of this approach to rotation-collocation algorithm comes from running an orbit propagator as implied for determination of $u(t)$ in equation 2a, which is executed $N$ times for each RO sounding. If there are $r$ total RO

soundings and $N$ sub-occultations per RO sounding, the time complexity of orbit propagation is $O(rN)$, and does not depend on the number of nadir scanner soundings. As a result, the rotation-collocation method is significantly faster than either brute-force method when there are large numbers of nadir scanner soundings.

### 2.2.2    Rotation-collocation #2: Linearized

In the linearized approach to the rotation-collocation algorithm, the positions of only two of the RO sub-occultations are
computed, and those are at $t = t_{\mathrm{RO}} - \Delta t$ and at $t = t_{\mathrm{RO}} + \Delta t$, and the line segment connecting those two positions in the rotated frame is checked for crossing the scan line. If it does cross the scan line ($\delta u = 0$), the temporal check is satisfied, and if it crosses the scan line at $|\delta s| < \delta s_{\max}$, then the spatial check is satisfied and a collocation is found.

    The computational expense of this approach to the rotation-collocation algorithm comes from running an orbit propagator as implied for determination of $u(t)$ in equation 2a, which is executed only 2 times for each RO sounding. As such, if there are $r$
total RO soundings, the time complexity of orbit propagation is $O(r)$. Recalling that the time complexity of orbit propagation is $O(rN)$ for the rotation-collocation algorithm with sub-occultations, when $N$ is much greater than 2, the linearized approach to collocation is much faster than the sub-occultation approach; however, it can be less accurate because the path of the RO sounding in the rotated frame is not strictly a straight line. The greater the temporal window $\Delta t$ is, the more curved the trajectory becomes. As explored in §4.6, as $\Delta t$ grows and the trajectory curvature increases, the number of incorrect predictions made
by the linearized rotation-collocation method also increases, and using the rotation-collocation method with sub-occultations becomes necessary to preserve accuracy.

## 3    Experimental setup

We devise a set of experiments to test the validity of the approximations of the rotation-collocation algorithm posed in the introduction and evaluate the computational efficiency gains for each. The experiments consist of a month of geolocations of
actual RO data and nadir-scanning data from January 2021. Because of the promise in using nadir microwave radiance to construct weather-independent temperature and water vapor profiles from the surface to the stratopause, we use the geolocations of highly precise, well-calibrated microwave nadir radiance data. The nadir-scanner geolocations are for the Advanced Microwave Sounding Units (AMSU-A) instruments on the Metop satellites (Metop-B and Metop-C) and for the Advanced Technology Microwave Sounders (ATMS) on the Suomi-NPP and NOAA-20 satellites. All are in sun-synchronous orbits with the Metop
satellites having their ascending node at 21:31 local solar time and the Suomi-NPP and NOAA-20 satellites at 13:25 local solar time. In January 2021, all four of these microwave radiance instruments collected 238,198,740 soundings, as detailed in Table 1.

**Table 1.** Total number of microwave radiance soundings over the month of January 2021 for each nadir-scanning microwave satellite.

|  | Number of MW soundings |
|---|---|
| **NOAA-20** | 121,070,688 |
| **Metop-B-AMSU** | 10,265,310 |
| **Metop-C-AMSU** | 10,111,350 |
| **SNPP** | 96,751,392 |
| **All** | 238,198,740 |

For the RO sounders, we choose two contemporary RO constellations: the two-satellite constellation of Metop consisting of Metop-B and Metop-C, and the six-satellite constellation of COSMIC-2. (Note that the Metop satellites carry both nadir microwave scanners and RO instruments.) These RO satellites are characterized by high signal-to-noise ratios for signal tracking but differ substantially in their orbits. The Metop satellites fly in sun-synchronous orbits, as above, while the COSMIC-2 satellites fly in $24°$ inclination, 520 km altitude, rapidly precessing orbits. In January 2021, the eight RO satellites obtained 160,298 RO soundings, as detailed in Table 2. By choosing these very different RO orbits, we not only can test the rotation-collocation algorithm, we also gain some insight into the frequency of microwave-RO collocations according to orbit types. Co-hosted instruments such as on the Metop satellites can intuitively be expected to yield greater numbers of collocations daily than RO and microwave radiance instruments on different satellites with unrelated orbits – about 40% of Metop RO soundings are collocated with Metop microwave radiance soundings, whereas generally under 5% of RO soundings are collocated with microwave radiance soundings from any instrument hosted on a different satellite in an unrelated orbit.

**Table 2.** Total number of RO soundings over the month of January 2021 for each RO satellite. Note that the COSMIC-2 constellation contains six satellites. No Metop-C-GRAS soundings are available for January 17, 2021, so there are fewer Metop-C-GRAS soundings than Metop-B-GRAS soundings.

|  | Number of RO soundings |
|---|---|
| **COSMIC-2** | 125,665 |
| **Metop-B-GRAS** | 18,140 |
| **Metop-C-GRAS** | 16,493 |
| **All** | 160,298 |

All computations are done in Python version 3.11. The orbit propagator used in computing $u(t)$, $i(t)$, and $\Omega(t)$ of equation 2a is *sgp4* (Vallado et al., 2006). It is initiated approximately three times daily from two-line orbit elements (TLEs). The transformation between ECF and ECI coordinate frames of equation 2b is executed using *astropy*, with the ECF chosen to be the International Terrestrial Reference System (ITRS) and the ECI frame the Geocentric Celestial Reference System (GCRS)

(Price-Whelan et al., 2022). The conversion is executed only three times for every two-line element description in the Celestrak database in order to establish the Earth's pole and rotation rate. Subsequent transformations between ECF and ECI are executed using the calculated pole and rotation rate.

We obtained the Metop data from Eumetsat (https://eoportal.eumetsat.int/), and the NOAA-20 and Suomi-NPP data from NOAA's CLASS data system (https://www.class.noaa.gov/). We retrieved the RO sounding data from the COSMIC Data Analysis and Archive Center (https://data.cosmic.ucar.edu/gnss-ro/). We also retrieved historical TLEs for Suomi-NPP, Metop-B, Metop-C, NOAA-20, and the COSMIC-2 constellation from Celestrak (https://celestrak.org/) for use in the rotation-collocation method. We grouped data into folders by instrument and day, and then ran all four methods on each combination of instruments per day.

## 4  Analysis

We analyze the performance of the two approaches of the rotation-algorithm using signal detection theory—counting false positive and false negative rates—using the brute force algorithm as the definition of truth. Because the two approaches to the brute force algorithm are provably the same despite their different approaches to checking for temporal match-ups, they both yield precisely the same collocation pairs. In this section we present a set of case studies. In each case we choose a spatial tolerance of $\Delta s = (150 \text{ km})/R_E$, and in all cases but the last we choose a time window of $\Delta t = 600 \text{ sec}$, or ten minutes; in the fourth and final case we choose a time window of $\Delta t = 10800 \text{ sec}$, or three hours. Occultation yield can be expected to increase in direct proportion to $\Delta t$ for time windows significantly shorter than the orbital period of the nadir scanning satellites. The first case study considers collocations between COSMIC-2 RO soundings and NOAA-20 microwave radiance soundings. This is a typical case since many future RO instruments will not necessarily be co-hosted with microwave radiance sounders and will be in different orbits. The second case study is for the co-hosted RO and microwave radiance soundings on the Metop satellites. While not many such pairings will be deployed in the future, it may suggest that RO and microwave radiance sounders be flown in tandem orbits if maximizing the collocation yield is desired. Third, the total yield of RO-microwave radiance collocations for the month of January 2021 is considered. The final case study reconsiders collocations between COSMIC-2 RO soundings and NOAA-20 microwave radiance soundings but with a time window of $\Delta t = 10800 \text{ sec}$. This final case study demonstrates the excellent accuracy and efficiency of the rotation-collocation method with sub-occultations over long time windows, and documents the slight decrease in accuracy of the linearized rotation-collocation method as the curvature of the trajectory of sub-occultations in the nadir-scanner frame increases over a longer time window.

 **4.1 Case study: COSMIC-2 (RO) and NOAA-20 (microwave)**

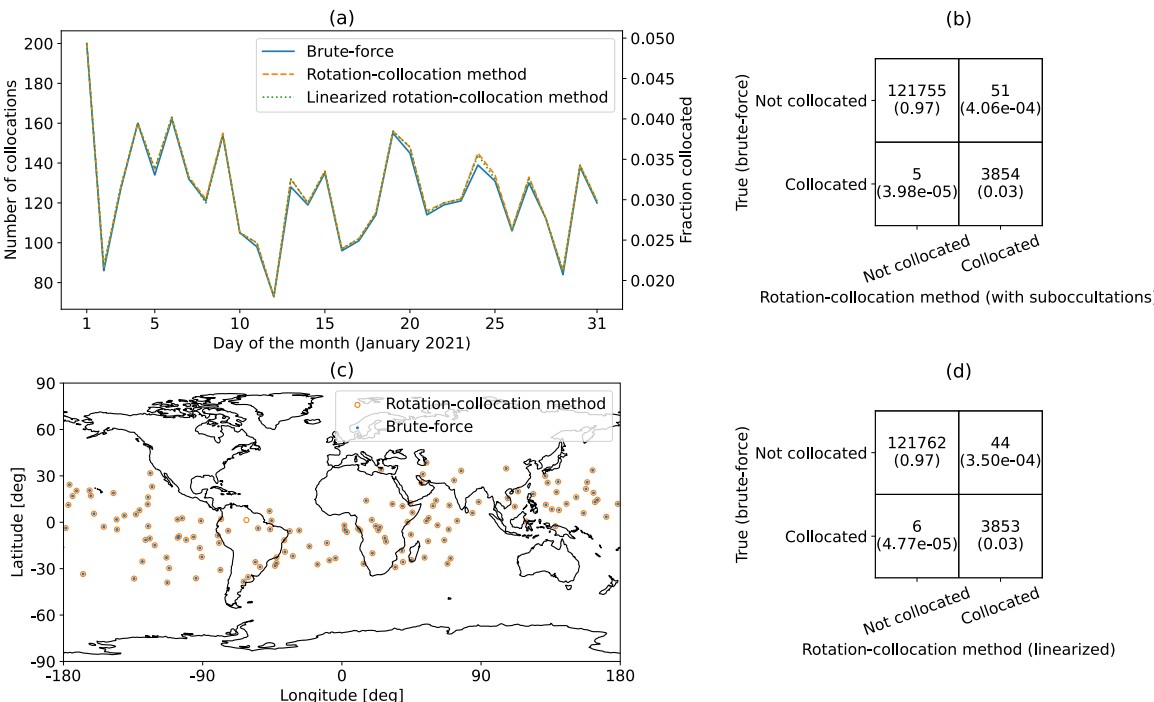

**Figure 2.** (a) Daily collocations for COSMIC-2 RO soundings and NOAA-20 ATMS microwave radiance soundings for the month of January 2021. (b) Confusion matrix for the COSMIC-2 RO soundings and NOAA-20 ATMS microwave soundings for January 2021 for the rotation-collocation method with sub-occultations. (c) Map of collocations on January 15, 2021, for NOAA-20 ATMS soundings and COSMIC-2 RO soundings. (d) Confusion matrix for the COSMIC-2 RO soundings and NOAA-20 ATMS microwave soundings for January 2021 for the linearized rotation-collocation method.

In this case study, we examine collocations between the six-satellite COSMIC-2 radio occultation constellation and ATMS on NOAA-20, a microwave radiance sounder. In Figure 2(a), we show the collocations between COSMIC-2 and NOAA-20 by day found by each of our four collocation-finding methods. Both brute-force methods yield identical results, and so both methods are represented in Figure 2(a) by the same blue line. The rotation-collocation algorithm with sub-occultations (orange)
and the linearized rotation-collocation algorithm (light green) find slightly more collocations on each day than the brute-force algorithms (blue), but the true positive rate, defined as the number of collocations correctly predicted by the rotation-collocation method divided by the total number of predicted collocations correctly or incorrectly, is over 98.5% for both versions of the rotation-collocation method. The time window for collocation for Figure 2 is $\Delta t = 600$ sec. The "fraction collocated" axis on the right of Figure 2(a) is the number of predicted collocations divided by the average number of daily occultations. Notably,
only 2% to 5% of COSMIC-2 RO soundings are collocated with NOAA-20 ATMS microwave radiance soundings over the

month of January 2021 when $\Delta t = 600$ is used as the time tolerance for collocation, because NOAA-20 and COSMIC-2 satellites are rarely near each other.

In Figure 2(b), we show a confusion matrix for this case study. The number of sub-occultations used for this analysis is $N = 21$, and the temporal spacing between sub-occultations is $dt = 60$ sec. In the confusion matrix, the top and bottom rows
correspond to the numbers of collocations of RO soundings not found and found by brute force, respectively; and the left and right columns to the numbers of collocations of RO soundings not predicted and predicted by one of the rotation-collocation methods, respectively. The true positive rate for the rotation-collocation method with sub-occultations is $3854/3905 = 98.7\%$ and the true negative rate is $121755/121760 = 99.996\%$.

In Figure 2(c), we show the spatial distribution of COSMIC-2 soundings collocated with NOAA-20 microwave radiance
soundings for January 15, 2021, found by the linearized rotation-collocation algorithm and by the brute-force method. Collocations found by the linearized rotation-collocation algorithm are shown as orange circles, and those found by the brute force algorithm are shown as blue dots. The vast majority of these collocated soundings are found by both methods. The brute force algorithm found 135 collocations, and the rotation-collocation algorithm found 136 collocations – the same 135 collocations found by the brute force algorithm, plus an extra collocation. For this day, the true positive rate is $135/136 = 99.3\%$.

In Figure 2(d), we show a confusion matrix for collocations found by the linearized rotation-collocation algorithm between COSMIC-2 RO soundings and NOAA-20 ATMS soundings for the month of January 2021. The linearized rotation-collocation algorithm finds the same collocations as the rotation-collocation method with sub-occultations in this case, and so the true positive rate for the linearized rotation-collocation method is $3853/3897 = 98.9\%$ and the true negative rate is $121762/121768 = 99.995\%$.

Many of the COSMIC-2 RO soundings misclassified by the linearized rotation-collocation method (44 out of 50 total) are incorrect predictions, predicting a collocation when one does not exist. We found that 7 (15.9% of total) are soundings that fall just outside the time window $\Delta t$. This occurs when one endpoint of the apparent RO scan pattern in the coordinate frame given by NOAA-20's orbit lies close to, but does not cross, the $\delta u = 0$ line. The remaining 37 false positives (84.1% of total) are soundings that fall just outside of the maximum scan range $\delta s$ of the NOAA-20 ATMS instrument. One such false positive
is pictured in Figure 3.

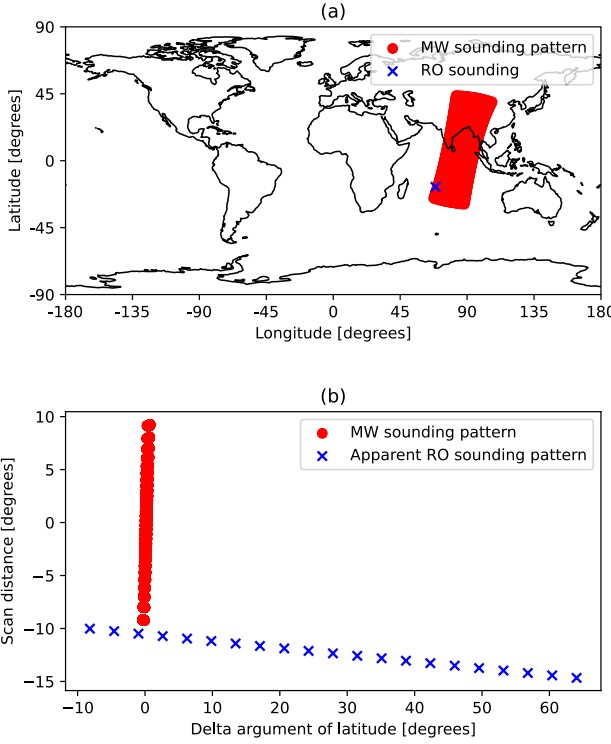

**Figure 3.** (a) A radio occultation from COSMIC-2, occurring at 7:55:12 AM GMT on January 3, 2021, falsely identified by the rotation-collocation method as a collocation, plotted against contemporaneous MW soundings from NOAA-20 ATMS. (b) The discretized apparent position of the same occultation rotated into the MW frame, plotted against the MW sounding pattern.

All of the false positive and false negative cases found here are associated with failures of the first assumption of the rotation-collocation algorithm, namely, that all of the nadir scanner soundings fall perfectly on an unbroken line at $\delta u = 0$ in the rotated frame as illustrated by Figure 1(c). There are more false positives than false negatives because of our windowing criteria, and adjusting these criteria would lead to more false negatives but fewer false positives. All the false positives and false negatives occur very close to the spatial or temporal boundaries for collocation, and so these misclassified soundings represent low-value collocations compared to other soundings that have more temporal and spatial overlap with the nadir-scanner sounding pattern.

In summary, the rotation-collocation algorithm with sub-occultations is correct on 98.7% of the occasions for which a collocation between COSMIC-2 RO soundings and NOAA-20 ATMS soundings is predicted and incorrect only 0.004% of the time when a COSMIC-2 RO sounding is not found to be collocated with a NOAA-20 ATMS sounding. The linearized rotation-collocation algorithm is correct on 98.9% of the occasions for which a collocation between COSMIC-2 RO soundings and NOAA-20 ATMS soundings is predicted and incorrect only 0.005% of the time when a COSMIC-2 RO sounding is not found to be collocated with a NOAA-20 ATMS sounding. Over the course of January 2021, the true number of collocated

soundings between COSMIC-2 RO and NOAA-20 ATMS soundings within a time window of 10 minutes is 3,859. The yield as a fraction of total COSMIC-2 RO soundings is 3.1% over the month. On a daily basis, the fraction ranges from 2.0% to 5.0%; see Figure 2(a).

## 4.2 Case study: Metop-B (RO) and Metop-B (microwave)

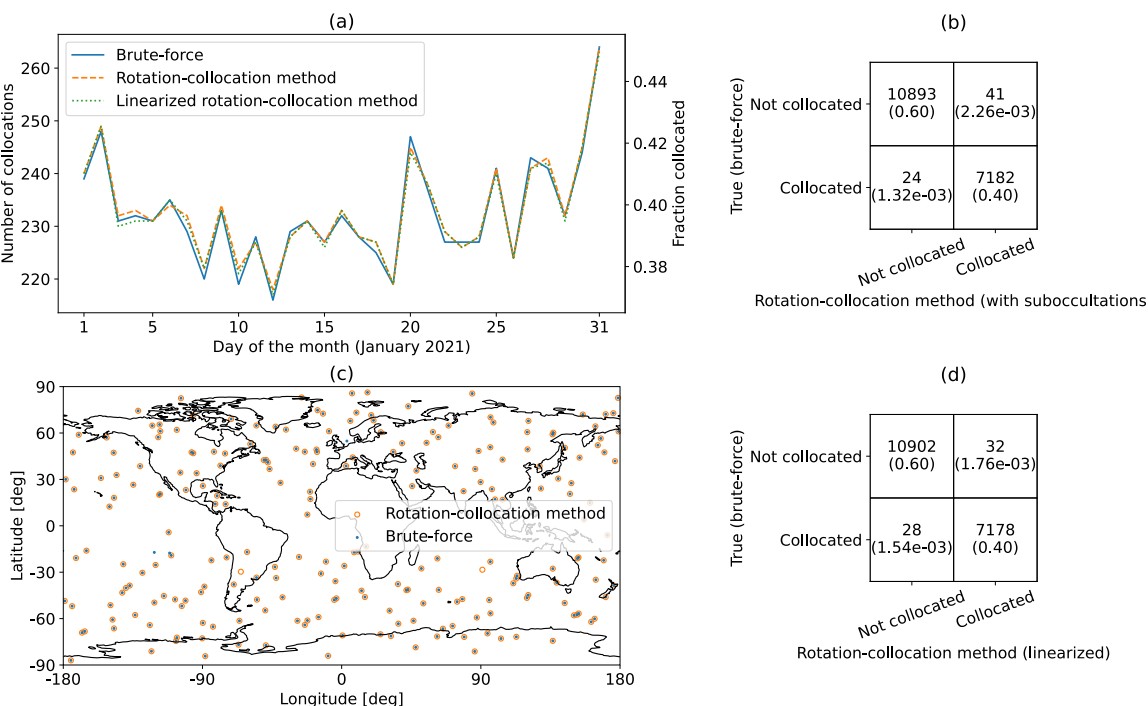

**Figure 4.** (a) Daily collocations for Metop-B GRAS RO soundings and Metop-B AMSU-A microwave radiance soundings for the month of January 2021. (b) Confusion matrix for the Metop-B GRAS RO soundings and Metop-B AMSU-A ATMS microwave soundings for January 2021 for the rotation-collocation method with sub-occultations. (c) Map of collocations on January 15, 2021, for Metop-B AMSU-A microwave soundings and Metop-B GRAS RO soundings. (d) Confusion matrix for the Metop-B GRAS RO soundings and Metop-B AMSU-A microwave soundings for January 2021 for the linearized rotation-collocation method.

In this case study, we examine collocations between two instruments cohosted on a satellite, the GRAS RO instrument and the AMSU-A nadir-scanning microwave radiance instrument. Figure 4 is the same as Figure 2 but for this case study.

Co-hosting instruments greatly increases the collocation yield, with around 38–46% of Metop-B RO soundings collocated with Metop-B microwave soundings, in comparison to around 3% of COSMIC-2 RO soundings collocated with NOAA-20 microwave soundings. The intuition for this is straightforward. If a setting RO sounding is obtained at a time $t_{\mathrm{RO}}$, then it is very likely that the satellite had flown over that same location earlier by $L/v_{\mathrm{leo}}$ in which $L$ is the limb distance for the RO sounding and $v_{\mathrm{leo}}$ is the low-Earth orbiting satellite's orbital velocity. Typically, $L \simeq 3000$ km and $v_{\mathrm{leo}} \simeq 7.5$ km s$^{-1}$,

meaning a collocated microwave radiance sounding may have been swept out by the scanner approximately 400 seconds prior. The temporal collocation check is always satisfied for co-hosted RO and nadir scanning instruments as long as the spatial window is greater than 400 seconds ($\Delta t > 400$ s). For the collocation to be found, though, the boresight angle of the RO sounding with respect to the satellite's velocity vector must be less than the angle corresponding to the sweep of the AMSU-A scan $\delta s_{\max}$ as viewed at limb distance $L$. Maximum boresight angles for RO instruments typically lie around $60°$, but the nadir scan of AMSU-A corresponds to a maximum boresight of approximately $27°$ at limb distance. As a consequence, instead of all RO soundings by Metop-B being collocated with a Metop-B microwave sounding, approximately only 40% are. This corresponds to the spatial check for collocation only being met 40% of the time.

Figure 4(b) shows the performance of the rotation-collocation method with sub-occultations on collocations between Metop-B-GRAS and Metop-B-AMSU throughout the month of January 2021, using 21 sub-occultations, or a sixty-second spacing between sub-occultations. The true positive rate for the rotation-collocation method with sub-occultations is $7182/7223 = 99.4\%$ and the true negative rate for the rotation-collocation method with sub-occultations is $10893/10917 = 99.8\%$.

Figure 4(d) shows the performance of the linearized rotation-collocation method on collocations between Metop-B-GRAS and Metop-B-AMSU throughout the month of January, 2021. Most of the Metop-B-GRAS soundings misclassified by the linearized rotation-collocation method are false positives. We have found that 2 of 32 (6.25%) of the false positives are due to unavailable Metop-B-AMSU data – for these predicted collocations, there is no Metop-B-AMSU sounding data available within eight seconds of the predicted collocation time. The remaining 30 (93.75% of total) false positives are soundings that fall just outside of the maximum scan range $\delta u_{\max}$ of the Metop-B-AMSU instrument. The true positive rate for the linearized rotation-collocation method is $7178/7210 = 99.6\%$, and excluding incorrect predictions that occur due to missing data, the true positive rate increases slightly to $7178/7208 = 99.6\%$. The true negative rate for the rotation-collocation method is $10902/10930 = 99.7\%$.

 **4.3 Full analysis: COSMIC-2 and Metop (RO); S-NPP, NOAA-20, Metop (MW)**

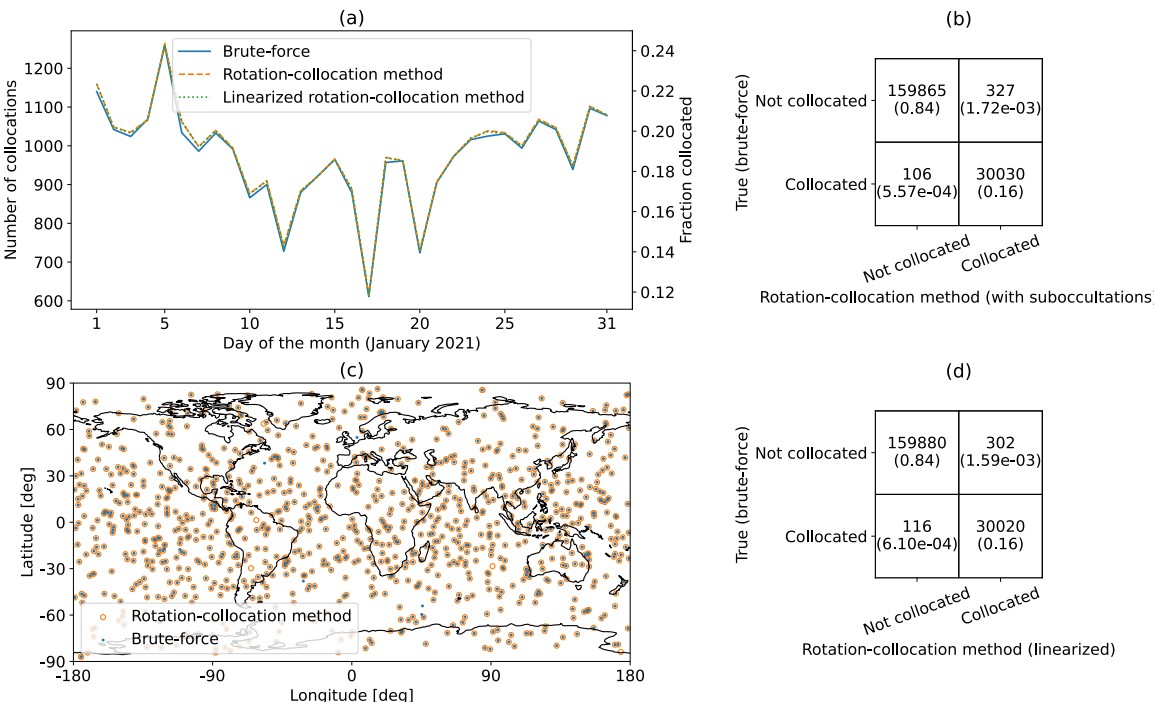

**Figure 5.** (a) Daily collocations for all satellite combinations for the month of January 2021. (b) Confusion matrix for all satellite combinations for January 2021 for the rotation-collocation method with sub-occultations. (c) Map of collocations on January 15, 2021, for all satellite combinations. (d) Confusion matrix for all satellite combinations for January 2021 for the linearized rotation-collocation method.

In this section, we examine collocations between COSMIC-2 and Metop-B and Metop-C radio occultations and Metop, S-NPP, and NOAA-20 microwave soundings. In Figure 5(a), we show the collocations by day found by each of our four collocation-finding methods, as well as the fraction of radio occultations that are collocated with microwave soundings, using the daily average number of radio occultations as the denominator. Metop-C-GRAS data was missing for January 17, which explains the steep drop in total collocations found on January 17. As before, the time window for collocation is $\Delta t = 600$s.

Over all satellite combinations, only 15.8% of RO soundings are collocated with any MW soundings; ideally, as many RO soundings would be collocated with MW soundings as possible. It is clear that there is room for improvement in the percentage of soundings that are collocated, and cohosting instruments leads to a large increase in collocations, as shown in §4.2.

In Figure 5(c), we show all the collocations on January 15, 2021. These collocations occur all over the globe. Collocations in the Tropics, between $23.43°$S and $23.43°$N in latitude, are most important for profiling water vapor in the planetary boundary layer (Wang et al., 2017). Future satellite missions with GNSS-RO payloads should consider cohosting microwave radiometer payloads or launching into low-inclination orbits in order to meet the need for collocations in the Tropics.

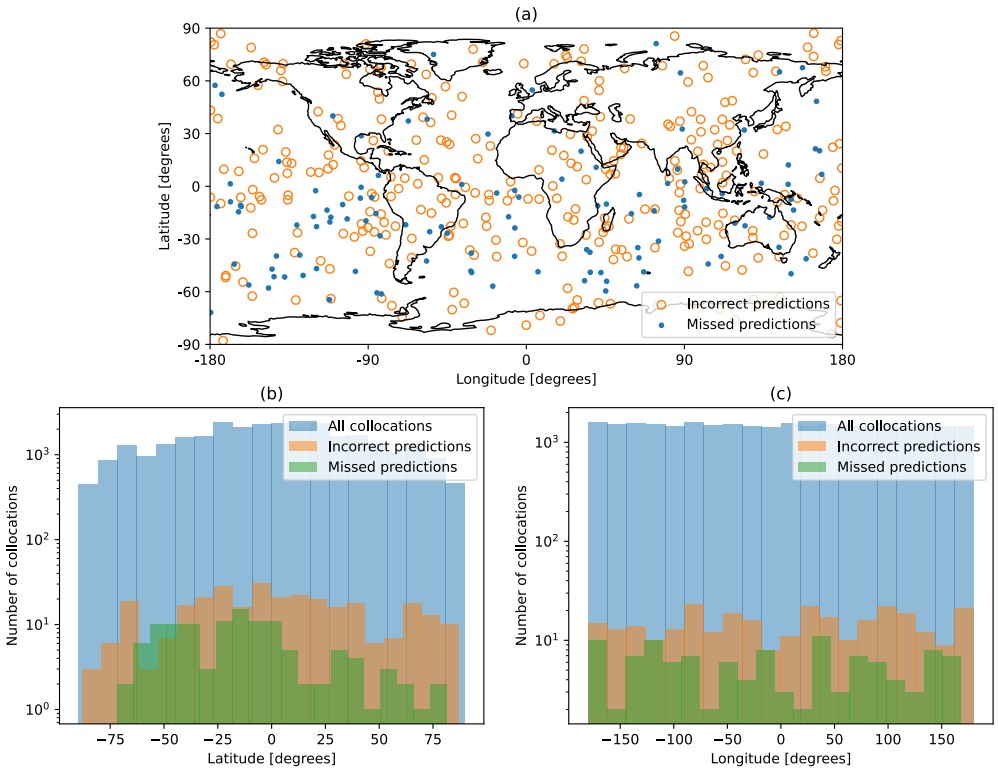

**Figure 6.** (a) Map of incorrect and missed predictions for all satellite combinations, (b) Histogram of latitude of all collocations, incorrect predictions, and missed predictions for all satellite combinations, (c) Histogram of longitude of all collocations, incorrect predictions, and missed predictions for all satellite combinations.

Overall, the linearized rotation-collocation method found 30,020 collocations and correctly identified 159,880 RO soundings as not collocated. There were 116 missed predictions, or occultations for which the brute-force method found a collocation but the linearized rotation-collocation method did not. There were 302 incorrect predictions, which are occultations where the linearized rotation-collocation method found a collocation, but the brute-force method did not. Out of these 302 incorrect predictions, 44 (14.6% of total) were caused by missing microwave data, 85 (28.1% of total) were soundings that fall just outside of the maximum scan range of an microwave instrument, and the remaining 173 (57.3% of total) were soundings that fall just beyond the maximum delta argument of latitude when compared to an MW satellite's orbit. The linearized rotation-collocation method had a $30020/30322 = 99.0\%$ true positivity rate and a $159880/159996 = 99.9\%$ true negative rate. Excluding incorrect predictions resulting from missing or corrupted microwave radiance data, the true positive rate is $30020/30278 = 99.1\%$.

Figure 6(a) shows the geographic distribution of incorrect predictions and missed predictions. Figures 6(b) and 6(c) display the distribution of latitude and longitude, respectively, for incorrect predictions, missed predictions, and all collocations. The set of all collocations is roughly centered at the equator and prime meridian, with a mean latitude of $0.49°$, mean longitude of $-1.69°$, standard deviation of latitude of $42.2°$, and standard deviation of longitude of $104.1°$. The distribution of incorrect predictions is similar, with a mean latitude of $2.82°$, mean longitude of $4.52°$, standard deviation of latitude of $42.2°$, and standard deviation of longitude of $103.4°$. The distribution of missed predictions, however, is centered slightly south of the equator; it has a mean latitude of $-12.58°$, mean longitude of $-10.5°$, standard deviation of latitude of $34.1°$, and standard deviation of longitude of $105.4°$. The sample size ($n = 116$) of missed predictions is small, however, which makes it difficult to evaluate the significance of this small shift in geographic distribution.

## 4.4 Number of collocations by day, by satellite combinations

Table 3 shows the number of collocations per day over the month of January 2021 by satellite combinations. Metop-B-AMSU and Metop-B-GRAS generate a large yield of collocations. These instruments are co-hosted, which allows a high percentage of occultations to be collocated with microwave soundings. For the same reason, Metop-C-AMSU and Metop-C-GRAS share a high number of collocations. There are no collocations between Metop-B-AMSU and Metop-C-GRAS, and none between Metop-C-AMSU and Metop-B-GRAS. Metop-B and Metop-C have co-planar orbits but are approximately half an orbit apart within their orbital plane. As such, their trajectories never intersect or get sufficiently close for measurements from their instruments to be collocated.

**Table 3.** Number of collocations by day, using $\Delta t = 600$ seconds as the temporal criterion and $\Delta d = 150$ km as the spatial criterion for collocation, by satellite combinations. The first row in each cell shows the average number of collocations per day found by both brute-force methods (recall that both brute-force methods yield an identical list of collocations), with the standard deviation of the number of collocations per day in parantheses. The second row shows the same metrics for the rotation-collocation method with sub-occultations, and the third shows the same metrics for the linearized rotation-collocation method.

|  | Collocations | NOAA-20 | Metop-B-AMSU | Metop-C-AMSU | SNPP | All |
|---|---|---|---|---|---|---|
| **COSMIC-2** | Brute-force | 124.5 (25.6) | 106.0 (24.5) | 100.4 (21.4) | 124.6 (28.2) | 455.5 (71.7) |
|  | Rot.-coll. w/ sub-occ | 126.0 (25.7) | 106.4 (24.5) | 101.7 (22.0) | 125.7 (28.0) | 459.8 (71.8) |
|  | Linear. rot.-coll. | 125.7 (25.6) | 106.4 (24.4) | 101.6 (22.1) | 125.6 (28.0) | 459.3 (71.9) |
| **Metop-B-GRAS** | Brute-force | 10.6 (17.0) | 232.5 (9.9) | 0.0 (0.0) | 26.7 (26.5) | 269.7 (26.4) |
|  | Rot.-coll. w/ sub-occ | 10.7 (17.2) | 233.0 (9.5) | 0.0 (0.0) | 27.1 (26.9) | 270.8 (26.4) |
|  | Linear. rot.-coll. | 10.7 (17.2) | 232.6 (9.5) | 0.0 (0.0) | 27.1 (26.9) | 270.4 (26.4) |
| **Metop-C-GRAS** | Brute-force | 23.2 (26.0) | 0.0 (0.0) | 218.0 (44.5) | 13.9 (20.0) | 246.9 (72.8) |
|  | Rot.-coll. w/ sub-occ | 23.3 (26.1) | 0.0 (0.0) | 219.4 (44.2) | 14.3 (20.7) | 248.7 (73.0) |
|  | Linear. rot.-coll. | 23.3 (26.1) | 0.0 (0.0) | 219.1 (44.1) | 14.3 (20.7) | 248.5 (73.0) |
| **All** | Brute-force | 157.5 (35.3) | 338.4 (27.0) | 311.4 (66.1) | 164.8 (36.0) | 972.1 (124.1) |
|  | Rot.-coll. w/ sub-occ | 159.2 (35.4) | 339.4 (26.9) | 314.0 (66.1) | 166.6 (36.2) | 979.3 (124.4) |
|  | Linear. rot.-coll. | 159.0 (35.3) | 339.0 (26.9) | 313.6 (66.2) | 166.6 (36.2) | 978.1 (124.4) |

## 4.5 Computational expense analysis and accuracy

Table 4 shows the core-minutes per day of RO data required to compute collocations for different combinations of satellites on an 8-core 2020 MacBook Pro with an M1 chip and 16 GB of RAM. The fastest method, the linearized rotation method, takes on average less than one core-minute per day to compute collocations for all satellites and achieves a 328-fold acceleration over the sorted brute-force method. The acceleration by the linearized rotation-collocation method varies depending on the time tolerance and computational hardware used but in general ranges between 40-fold and 400-fold over conventional brute

force algorithms.

**Table 4.** Core-minutes required for computation by day, using $\Delta t = 600$ seconds as the temporal criterion and $\Delta d = 150$ km as the spatial criterion for collocation, by satellite combinations (excluding data-loading). The first row in each cell shows the average core-minutes required to compute the collocations for a satellite pair for a single day using brute-force method #1, with the standard deviation of core-minutes taken for computation time in parentheses. The second row shows the same metrics for the sorted brute-force method, the third shows the same metrics for the rotation method with sub-occultations, and the fourth shows the same metrics for the linearized rotation-collocation method.

|  | Collocations | NOAA-20 | Metop-B-AMSU | Metop-C-AMSU | SNPP | All |
|---|---|---|---|---|---|---|
| **COSMIC-2** | Brute-force #1 | 13.1 (1.8) | 7.3 (0.9) | 7.4 (1.3) | 35.7 (4.9) | 63.5 (8.7) |
|  | Brute-force #2 | 11.6 (1.5) | 2.6 (0.3) | 4.0 (8.2) | 33.6 (4.5) | 51.7 (11.6) |
|  | Rot.-coll. w/ sub-occ | 0.3 (0.0) | 0.3 (0.0) | 0.3 (0.0) | 0.3 (0.0) | 1.3 (0.2) |
|  | Linear. rot.-coll. | 0.0 (0.0) | 0.0 (0.0) | 0.0 (0.0) | 0.0 (0.0) | 0.2 (0.0) |
| **Metop-B-GRAS** | Brute-force #1 | 1.9 (0.0) | 0.9 (0.0) | 1.1 (0.1) | 5.2 (0.2) | 9.1 (0.3) |
|  | Brute-force #2 | 1.7 (0.0) | 0.3 (0.0) | 0.4 (0.0) | 4.9 (0.2) | 7.3 (0.2) |
|  | Rot.-coll. w/ sub-occ | 0.0 (0.0) | 0.0 (0.0) | 0.0 (0.0) | 0.0 (0.0) | 0.2 (0.0) |
|  | Linear. rot.-coll. | 0.0 (0.0) | 0.0 (0.0) | 0.0 (0.0) | 0.0 (0.0) | 0.0 (0.0) |
| **Metop-C-GRAS** | Brute-force #1 | 1.8 (0.4) | 1.0 (0.2) | 0.9 (0.2) | 4.8 (0.9) | 8.2 (2.2) |
|  | Brute-force #2 | 1.6 (0.3) | 0.4 (0.1) | 0.3 (0.1) | 4.6 (0.9) | 6.6 (1.8) |
|  | Rot.-coll. w/ sub-occ | 0.0 (0.0) | 0.0 (0.0) | 0.0 (0.0) | 0.0 (0.0) | 0.2 (0.0) |
|  | Linear. rot.-coll. | 0.0 (0.0) | 0.0 (0.0) | 0.0 (0.0) | 0.0 (0.0) | 0.0 (0.0) |
| **All** | Brute-force #1 | 16.8 (1.9) | 9.1 (1.0) | 9.4 (1.4) | 45.5 (5.4) | 80.8 (9.6) |
|  | Brute-force #2 | 14.8 (1.7) | 3.2 (0.4) | 4.7 (8.2) | 42.9 (5.0) | 65.6 (12.1) |
|  | Rot.-coll. w/ sub-occ | 0.4 (0.0) | 0.4 (0.0) | 0.4 (0.0) | 0.4 (0.0) | 1.6 (0.2) |
|  | Linear. rot.-coll. | 0.1 (0.0) | 0.1 (0.0) | 0.1 (0.0) | 0.1 (0.0) | 0.2 (0.0) |

## 4.6 Longer timescale analysis

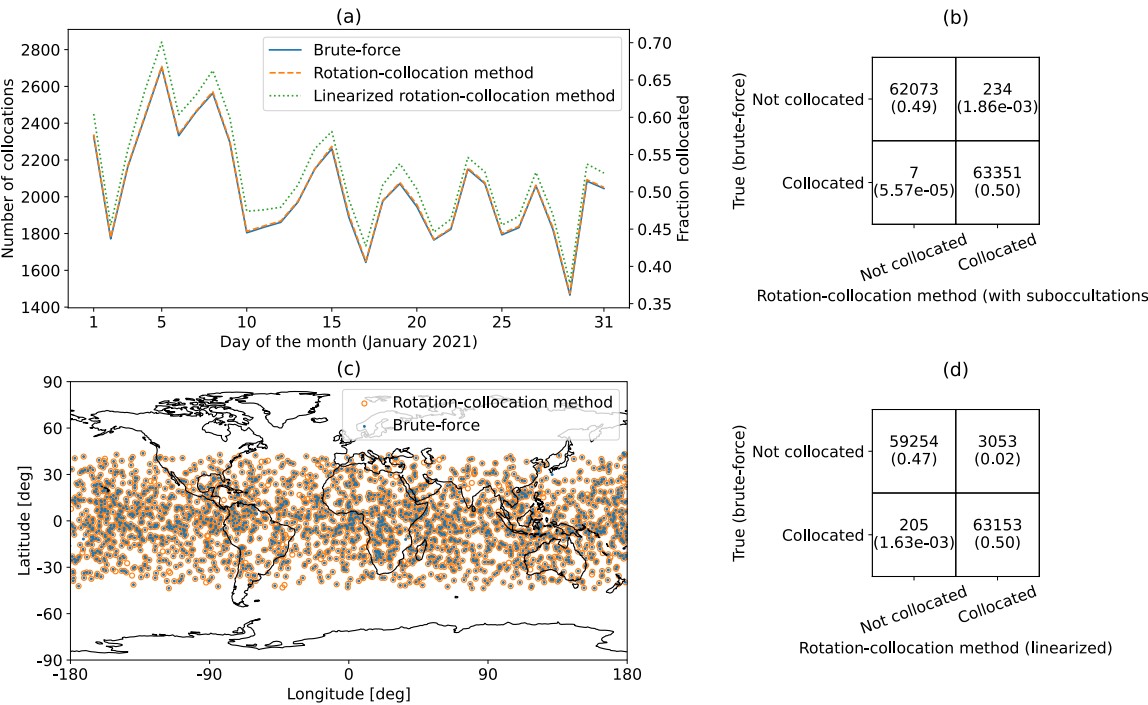

**Figure 7.** (a) Daily collocations for COSMIC-2 RO soundings and NOAA-20 ATMS microwave radiance soundings for the month of January 2021, with a 3-hour time tolerance. (b) Confusion matrix for the COSMIC-2 RO soundings and NOAA-20 ATMS microwave soundings for January 2021 for the rotation-collocation method with sub-occultations. (c) Map of collocations on January 15, 2021, for NOAA-20 ATMS soundings and COSMIC-2 RO soundings. (d) Confusion matrix for the COSMIC-2 RO soundings and NOAA-20 ATMS microwave soundings for January 2021 for the linearized rotation-collocation method.

Recall the second assumption outlined in §2.2: that the apparent position of an RO sounding in a nadir-sounder frame forms a linear trajectory. Over longer timescales, this trajectory elongates and its curvature becomes more apparent. To test the validity of this assumption, we applied all four collocation-finding methods to finding collocations between NOAA-20 ATMS and

COSMIC-2 with $\Delta t = 3 \text{ hours}$, a time window 18 times longer than that used for §4.1–§4.5. Increasing the time tolerance in this way greatly increases the number of possible collocations. For the rotation-collocation method with sub-occultations, we used $N = 5$ sub-occultations, or a spacing of $dt = 5400 \text{ s}$ between sub-occultations.

Figure 7(a) shows the collocations by day on the left vertical axis and fractional yield of collocations on the right vertical axis for NOAA-20 and COSMIC-2 over January 2021. With $\Delta t = 10800 \text{ s}$ (3 hours), the linearized rotation-collocation method

(light green) finds many more collocations than the brute-force algorithm (blue) and the rotation-collocation method with sub-occultations (orange). It is also apparent that with a time window of 3 hours, around half of all COSMIC-2 RO soundings are collocated with NOAA-20 soundings, many more than with a time window of 10 minutes.

**Table 5.** Total number of incorrect predictions (collocations identified by rotation-collocation method but not by the brute-force method), total number of missed predictions (collections missed by rotation-collocation method but found by the brute-force method), and total number of correct predictions (collocations found by both methods) for collocations between NOAA-20 ATMS soundings and COSMIC-2 RO soundings over the month of January 2021 with a 3-hour time tolerance for collocation, for the rotation-collocation method evaluated with a varying number of sub-occultations.

| Number of sub-occultations | Time between sub-occultations | Incorrect pre-dictions | Missed predic-tions | Correct pre-dictions |
|---|---|---|---|---|
| 2 (same as linearized) | 6 hours | 3053 | 205 | 63153 |
| 3 | 3 hours | 481 | 149 | 63209 |
| 4 | 2 hours | 282 | 86 | 63272 |
| 5 | 90 minutes | 234 | 7 | 63351 |
| 6 | 72 minutes | 229 | 10 | 63348 |
| 7 | 60 minutes | 225 | 9 | 63349 |

Figure 7(b) shows the performance of the rotation-collocation method with sub-occultations on collocations between COSMIC-2 and NOAA-20 for January, 2021. This method has a true positive rate of $63351/63585 = 99.6\%$ and a true negative rate of $62073/62080 = 99.99\%$. Figure 7(d) shows the performance of the linearized rotation-collocation method on collocations between COSMIC-2 and NOAA-20 for January 2021. This method has a true positive rate of $63153/66206 = 95.4\%$ and a true negative rate of $59254/59459 = 99.7\%$.

Although the linearized rotation-collocation method has many more incorrect predictions than the rotation-collocation method with sub-occultations, it retains a $95.4\%$ true positive rate. This illustrates that even over a three-hour period, the linearization of the trajectory of the apparent RO sounding in the nadir sounder frame is good enough to maintain a high level of accuracy. Also notable is that the sub-occultations used in this case study are spaced ninety minutes apart, longer than the twenty-minute spacing between endpoints used by the linearized rotation method in §4.1–§4.5. Even so, with a ninety-minute spacing between sub-occultations, there is a true positive rate of $99.9\%$ and only 234 incorrect predictions and 7 missed predictions for collocations between NOAA-20 and COSMIC-2 over the month of January 2021, which is better than the true positive rate of $98.9\%$ found with a twenty-minute spacing between sub-occultations for collocations between NOAA-20 and COSMIC-2 in §4.1. A ninety-minute spacing between sub-occultations is sufficient to achieve the accuracy demonstrated in sections §4.1–§4.5; longer time windows between sub-occultations result in more incorrect and missed predictions and reduced accuracy, as demonstrated in Table 5. The correlation between time between sub-occultations and accuracy breaks down as sub-occultations get close enough in time that the trajectory of the apparent RO sounding in the nadir sounder frame becomes approximately linear, at which point adding sub-occultations increases computation time without improving performance. This phenomenon can be seen in Table 5 – accuracy greatly improves as more sub-occultations are added, up to $N = 5$ sub-occultations, after which point performance remains relatively consistent.

Even with $\Delta t = 3$ hours, the rotation-collocation method remains extremely fast. On average, the brute-force method took 156.2 core-minutes to compute collocations for a single day of COSMIC-2 RO data, and the sorted brute-force method took 155.5 core-minutes to compute a day's worth of collocations. In contrast, the rotation-collocation method with sub-occultations took just 0.09 core-minutes to compute a day's worth of collocations and the linearized rotation-collocation method took 0.05 core-minutes on average to compute a day's worth of collocations. This results in a 3124-fold acceleration by the linearized rotation-collocation method over the brute-force method and a 1735-fold acceleration by the rotation-collocation method with sub-occultations over the brute-force method.

The apparent computational efficiency gains come about because the brute-force methods are decelerated more rapidly than $(\Delta t)^{-1}$ with longer time tolerance $\Delta t$. Brute-force methods only do the spatial check for nadir scan soundings that match in time, and so when many more soundings match in time, many more spatial checks are performed, which can be quite slow. This problem is particularly acute for the sorted brute-force method, which is actually the slowest method for a time window of 3 hours. The key advantage of the sorted brute-force method is that it considers many fewer nadir scanner soundings for each RO sounding than brute-force method #1 does. When the time window is long, this advantage evaporates but the time taken to search for the start and end of the time window in the sorted list of soundings remains, making the time taken by the sorted brute-force method similar to that taken by the brute-force method #1.

Additionally, because some RO soundings may occur at the very beginning or very end of a day, the brute-force methods must consider 30 hours of nadir scanner sounding, beginning 3 hours before the start of the day and ending 3 hours after the end of day, in order to find all collocations for a single day. With a 10-minute time tolerance for collocations, the brute-force methods only need consider 24 hours and 20 minutes of microwave soundings, speeding up the search for collocations. As a result, the acceleration provided by the rotation-collocation method is much more dramatic with $\Delta t = 3$ hours than with $\Delta t = 10$ minutes.

In conclusion, the rotation-collocation method retains remarkable accuracy when the time spacing between sub-occultations is 90 minutes or less. Even with a 3-hour spacing between sub-occultations, the rotation-collocation method retains accuracy above 95%. The time taken by the rotation-collocation method only scales with number of RO soundings and number of sub-occultations, whereas the time taken by the brute-force method scales with time tolerance. This makes the rotation-collocation method an excellent choice for finding collocations with time tolerances of 3 hours or more.

## 5   Conclusions

The rotation-collocation method has great potential to quickly find collocations between RO soundings and nadir-scan soundings. In fact, the rotation-collocation method generalizes easily, and can be applied to any set of sparsely sampled satellite data and any set of continuously sampled data from a nadir-scanning satellite. When applied to a month's worth of RO soundings from COSMIC-2, Metop-B-GRAS, and Metop-C-GRAS and a month's worth of MW soundings from Metop-B-AMSU, Metop-C-AMSU, SNPP, and NOAA-20 with a time tolerance of ten minutes, the linearized rotation-collocation method finds 30020 collocations with a 99.0% true positive rate and a 99.9% true negative rate and has a 328-fold acceleration over the

brute-force method. Furthermore, when incorrect predictions that result from missing microwave are held out, the linearized rotation-collocation method achieves a true positive rate of 99.1%. This indicates that when the time tolerance for collocation is low, the linearized rotation-collocation method achieves near perfect accuracy and does so hundreds of times faster than the fastest brute-force method.

When applied to a months' worth of COSMIC-2 RO soundings and NOAA-20 microwave soundings with a three-hour time tolerance for collocation, the rotation-collocation method with sub-occultations spaced ninety minutes apart achieves 99.6% true positive and 99.9% true negative rates with a 1735-fold acceleration over the fastest brute-force method. The linearized rotation-collocation method achieves 95.4% true positive and 99.6% true negative with a 3124-fold acceleration over the brute-force method. This demonstrates that the rotation-collocation method maintains a near perfect accuracy with sub-occultations

up to an hour apart, and that the rotation-collocation methods offer an improvement in speed over brute-force methods as the time tolerance for collocation is increased.

Currently, the geographic distribution of the soundings misclassified by the rotation-collocation algorithm roughly matches the geographic distribution of collocated soundings, as shown in Figure 6. Furthermore, most misclassified soundings are incorrect predictions (collocations predicted by the rotation-collocation algorithm but not by the brute-force method). Incorrect

predictions can be easily debunked, as the rotation-collocation algorithm currently predicts the expected time and scan angle of the collocated nadir-scanner sounding for each collocation, and it is computationally trivial to check if a real nadir-scanner sounding exists at the expected time and scan angle.

Finally, the rotation-collocation method shows that with a ten-minute time tolerance and 150 km spatial tolerance, there were an average of nearly 1000 collocated RO soundings each day of January 2021, or around 16% of all unique RO soundings

from Metop-B-GRAS, Metop-C-GRAS, and COSMIC-2. Around 40% of Metop-B-GRAS soundings were collocated with Metop-B-AMSU soundings, and around 40% of Metop-C-GRAS soundings were collocated with Metop-C-AMSU soundings. Cohosted instruments on Metop-B and Metop-C greatly increase the percentage of soundings that are collocated, and cohosting MW and RO instruments is a powerful tool for increasing the number of collocations.

## 5.1   Future Work and Applications

At present, the rotation-collocation algorithm identifies RO soundings which are collocated with nadir-scanner soundings, and additionally identifies the expected time and scan angle of the presumably-collocated nadir-scanner sounding. However, the rotation-collocation algorithm does not verify the existence of a nadir-scanner sounding at the expected time and scan angle, and so does not extract the specific nadir-scanner soundings associated with each collocation. The brute-force algorithms implemented in this paper also do not identify the specific nadir-scanner soundings associated with each collocation. In the

future, the authors plan to extend the rotation-collocation algorithm to identify the specific nadir-scanner soundings associated with each collocated RO sounding, and to integrate this extended version of the rotation-collocation algorithm into NASA's existing earth science data management software in order to speed up collocation-finding and assimilation of RO data into numerical weather prediction models.

The authors anticipate that extracting specific nadir-scanner soundings associated with each collocation will slow down both the rotation-collocation and brute-force methods, but will narrow the performance gap between the rotation-collocation and brute-force methods. Nevertheless, the authors expect that the rotation-collocation method will remain much faster than equivalent brute-force methods. The authors also plan to further investigate the geographic distribution of collocations missed by the rotation-collocation method.

The rotation-collocation method can be easily modified to identify collocations between two different nadir-scanning satellites. It can also be extended to predict collocation yield for satellite missions with nadir-scanning payloads in different orbits. In this way, the rotation-collocation method can be used as a constellation planning tool and a mission planning tool in order to select collocation-maximizing orbits for nadir-scanning satellites.

*Code and data availability.* The code associated with this paper will be made available at https://github.com/alexmeredith8299/ro-nadir-collocation. All of the data associated with this paper is freely available from UCAR, NOAA, and EUMETSAT. The code repository contains instructions on how to download the data.

*Author contributions.* AM performed data analysis and contributed to the manuscript. SL designed the study, provided advice on data analysis, and contributed to the manuscript. LH performed data analysis. KC provided advice on data analysis and on the manuscript. All authors contributed to the interpretation of the results and edited the manuscript.

*Competing interests.* The authors declare that they have no conflict of interest.

*Disclaimer.* Any opinion, findings, and conclusions or recommendations expressed in this material are those of the authors(s) and do not necessarily reflect the views of the National Science Foundation.

*Acknowledgements.* This work was supported by the National Science Foundation under grant GEO-1850089, the NASA Decadal Survey Incubator Program under grant 80NSSC22K1103, and the NSF Graduate Research Fellowship Program under Grant No. 2141064.

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
