# Peer review of "Efficient Collocation of GNSS Radio Occultation Soundings with Passive Nadir Microwave Soundings"

_EGUsphere, 2022_

## Author Comment (AC1)

Thank you so much for reviewing our paper! We've copied your review here in blue. We've responded to each part of your review separately, in black. We've also printed our planned revisions in response to your comments here, in black italics.

Review 1:

Combining radio occultation (RO) and passive microwave radiometer (MWR) is beneficial in many aspects. High vertical (RO - limb sounding) and horizontal (MWR – nadir sounding) resolution, RO bias correction, and MWR data calibration all make this combination appealing. However, searching for the collocation between RO and MWR is a time-consuming and computationally expensive task due to large number of MWR footprints and the random nature of the RO sounding locations. In this manuscript, authors developed a new algorithm called "rotation-collocation" method which significantly reduce the computation resources and time required for collocation identification. Based on its importance toward future mission and data analysis, I recommend this article to be published after minor revision.

One thing I would like to point out is that the comparison throughout the article with the traditional brute-force method is not entirely fair. The brute-force method gives us the pairs of footprints and RO location that satisfy the criteria, and the exact time and distance difference between the pairs. This information may either be missing after the coordinate transformation or needs further processing (which comes with extra computing complexity) using the new method. In addition to the accuracy and computing performance, I suggest to compare the final products between the two approaches as well.

This is a good point. However, the brute-force implementation used for this paper doesn't actually give us MW footprints that satisfy the criteria. Instead of checking all MW footprints satisfying the temporal match criterion, our implementation terminates early as soon as a collocation is found, which yields significant time savings when a high percentage of RO soundings are collocated and when the temporal criterion is lax (e.g. the $\Delta t$ = 3 hours case presented in section 4.6).

Furthermore, we have updated the rotation-collocation method to return predicted MW footprint time and scan angle for all collocated RO soundings (note that it still does not extract real collocated MW footprints). This change is relatively minor, as the rotation-collocation method was already internally calculating predicted MW footprint time, and so it has not had much effect on the computational resource consumption of the rotation-collocation method.

Extracting MW footprints from both methods is outside the scope of this paper, but we expect that full MW footprint extraction will not significantly alter the performance gap between the two methods. We do expect that MW footprint extraction will slightly slow down both methods.

We will explicitly state upfront in the introduction that we are not extracting MW footprints from any method in this paper, and we will also elaborate on our plans to extract MW footprints in the future in section 5.1.

Original lines 52-53: "*The algorithm for collocation involving rotation into the reference frame of the nadir scan pattern we refer to as the rotation-collocation method.*"

Revised lines 54-61: "*The algorithm for collocation involving rotation into the reference frame of the nadir scan pattern we refer to as the rotation-collocation method.*

*The rotation-collocation method implemented in this paper identifies RO soundings which cross the nadir scanner's scan line and predicts the approximate time and location of the closest nadir-scanner footprint to these RO soundings, but does not extract the nadir-scanner footprints collocated with these RO soundings. In order to fairly compare the rotation-collocation method to brute force methods, the brute force methods implemented in this paper also do not extract the nadir-scanner footprints associated with collocated RO soundings, and instead leverage early termination once a collocation is found for faster collocation-finding.*"

Original lines 436-441: "*At present, the rotation-collocation algorithm identifies RO soundings which are collocated with nadir-scanner soundings, but does not identify the specific nadir-scanner soundings associated with each collocation. In the future, the authors plan to extend the rotation-collocation algorithm to identify the specific nadir-scanner soundings associated with each collocated RO sounding, and to integrate this extended version of the rotation-collocation algorithm into NASA's existing earth science data management software in order to speed up collocation-finding and assimilation of RO data into numerical weather prediction models.*"

Revised lines 480-492: "*At present, the rotation-collocation algorithm identifies RO soundings which are collocated with nadir-scanner soundings, and additionally identifies the expected time and scan angle of the presumably-collocated nadir-scanner sounding. However, the rotation-collocation algorithm does not verify the existence of a nadir-scanner sounding at the expected time and scan angle, and so does not extract the specific nadir-scanner soundings associated with each collocation. The brute-force algorithms implemented in this paper also do not identify the specific nadir-scanner soundings associated with each collocation. In the future, the authors plan to extend the rotation-collocation algorithm to identify the specific nadir-scanner soundings associated with each collocated RO sounding, and to integrate this extended version of the rotation-collocation algorithm into NASA's existing earth science data management software in order to speed up collocation-finding and assimilation of RO data into numerical weather prediction models.*

*The authors anticipate that extracting specific nadir-scanner soundings associated with each collocation will slow down both the rotation-collocation and brute-force methods, but will narrow the performance gap between the rotation-collocation and brute-force methods. Nevertheless, the authors expect that the rotation-collocation method will remain much faster than equivalent brute-force methods.*"

L281: Not sure if I understand this sentence correctly. What is $\delta u_{max}$? It is the first time being mentioned in the manuscript without being defined. If it is a range of $\delta u$ like $\delta s_{max}$, how can a case fall beyond the range but simultaneously cross the scan line?

We will rephrase this sentence to avoid the use of delta u max, and at the end of the paragraph, we will explain the possible causes of these false positives (see our edits to line 285).

The idea here is that false positives fall into three categories: false positives due to data unavailability (i.e. a collocation would be present if nadir-scanner data was available), false positives at the temporal boundaries for collocation (i.e. one endpoint of the apparent RO scan pattern nearly hits, but does not quite cross, the $\delta u = 0$ line), and false positives at the spatial boundaries for collocation (i.e. the RO scan pattern crosses the $\delta u = 0$ line, but at a scan distance slightly greater in magnitude than the maximum nadir-scanner scan distance).

In this sentence, we are referring to false positives in the second of these three categories, which may be spatially collocated with MW soundings, but narrowly miss the criteria for temporal collocation.

The two major possible causes of this are orbit propagation error and the limitations of the approximation of the MW nadir scanner as continuously scanning. In both cases, the rotation-collocation method identifies the RO sounding as crossing the $\delta u = 0$ line, but this doesn't happen in the real world, because the rotation-collocation method simulates either the position or the footprints of the MW nadir scanner incorrectly.

Qualitatively, these RO soundings are edge cases at the temporal boundary for collocation, meaning that these soundings are not high-quality matchups, regardless of whether or not they strictly meet the criteria for collocation or are identified as collocations.

Original lines 280-282: *"Of the remaining false positives, 11 (25% of total) are soundings that fall just beyond the maximum $\delta u_{max}$ when compared to NOAA-20's orbit, thereby falling just outside the time window $\Delta t$."*

Revised lines 296-298: *"We found that 7 (15.9\% of total) are soundings that fall just outside the time window $\Delta t$. This occurs when one endpoint of the apparent RO scan pattern in the coordinate frame given by NOAA-20's orbit lies close to, but does not cross, the $\delta u = 0$ line."*

L285: If under the page limit, I think it would be great if a false positive and/or false negative case can be shown using Fig. 1(b) to illustrate the statement. Also, why is the number of false positive cases always larger than the one of false negative?

Yes, adding a figure to demonstrate what false positives look like is a great idea. We added a figure in the style of Fig 1(a) & Fig 1(b) showing a real false positive: a COSMIC-2 RO sounding which narrowly falls outside of the spatial boundaries for collocation.

We also added some text explaining why there are more false positives than false negatives -- it comes down to how exactly we draw the temporal and spatial boundaries for collocation, because false positives and false negatives occur at these boundaries. We also added further clarification of the fact that false positives and false negatives are edge cases that don't generally represent high-quality match-ups. Finally, later in the paper, we explain why we prefer

windowing criteria that yield more false positives than false negatives -- false positives are easy to debunk, but false negatives present a pure loss of information.

New figure:

[Figure]

**Figure 3.** (a) A radio occultation from COSMIC-2, occurring at 7:55:12 AM GMT on January 3, 2021, falsely identified by the rotation-collocation method as a collocation, plotted against contemporaneous MW soundings from NOAA-20 ATMS. (b) The discretized apparent position of the same occultation rotated into the MW frame, plotted against the MW sounding pattern.

Original lines 283-285: *"All of the false positive and false negative cases found here are associated with failures of the first assumption of the rotation-collocation algorithm, namely, that all of the nadir scanner soundings fall perfectly on an unbroken line at δu = 0 in the rotated frame as illustrated by Figure 1(c)."*

Revised lines 301-306: *"All of the false positive and false negative cases found here are associated with failures of the first assumption of the rotation-collocation algorithm, namely, that all of the nadir scanner soundings fall perfectly on an unbroken line at δu = 0 in the rotated frame as illustrated by Figure 1(c). There are more false positives than false negatives because of our windowing criteria, and adjusting these criteria would lead to more false negatives but fewer false positives. All the false positives and false negatives occur very close to the spatial or temporal boundaries for collocation, and so these misclassified soundings represent low-value collocations compared to other soundings that have more temporal and spatial overlap with the nadir-scanner sounding pattern."*

Original line 429: *"Finally, the rotation-collocation method shows that..."*

Revised lines 468-473: "*Furthermore, most misclassified soundings are incorrect predictions (collocations predicted by the rotation-collocation algorithm but not by the brute-force method). Incorrect predictions can be easily debunked, as the rotation-collocation algorithm currently predicts the expected time and scan angle of the collocated nadir-scanner sounding for each collocation, and it is computationally trivial to check if a real nadir-scanner sounding exists at the expected time and scan angle.*

*Finally, the rotation-collocation method shows that...*"

Fig 4 & 5: Maybe using the same format as Fig 3 and provide the confusion matrices?

Yes, we've replaced Figures 4 and 5 with a single figure in the same format as Figure 3, which includes the confusion matrices.

New figure:

[Figure]

**Figure 5.** (a) Daily collocations for all satellite combinations for the month of January 2021. (b) Confusion matrix for all satellite combinations for January 2021 for the rotation-collocation method with sub-occultations. (c) Map of collocations on January 15, 2021, for all satellite combinations. (d) Confusion matrix for all satellite combinations for January 2021 for the linearized rotation-collocation method.

Table 5: The number of sub-occultations (N) is negatively related to the prediction errors as expected. Can we observe the similar trend for previous cases (\Delta t = 600 s)? If so, the number of false predictions could also, at least partially, come from the nonlinearity of the RO curve instead of the \delta u=0 straight line assumption violation.

This trend doesn't generally hold for delta t = 600 s, because there's very little curvature on that timescale. For some satellite combinations (e.g. COSMIC-2/NOAA-20 as discussed in 4.1), the collocations found by the linearized rotation-collocation method and the rotation-collocation method with sub-occultations are identical. For others, (e.g. Metop-B/Metop-B as discussed in

4.2), different collocations are found by the two methods, but the true positive and false positive rates found by both methods are approximately the same.

We will make this explicit to the reader (when discussing table 5).

Original lines 383-385: *"A sixty-minute spacing between sub-occultations is sufficient to achieve the accuracy demonstrated in sections 4.1-4.5; longer time windows between sub-occultations result in more incorrect and missed predictions and reduced accuracy, as demonstrated in Table 5."*

Revised lines 416-422: *"A ninety-minute spacing between sub-occultations is sufficient to achieve the accuracy demonstrated in sections 4.1-4.5; longer time windows between sub-occultations result in more incorrect and missed predictions and reduced accuracy, as demonstrated in Table 5. The correlation between time between sub-occultations and accuracy breaks down as sub-occultations get close enough in time that the trajectory of the apparent RO sounding in the nadir sounder frame becomes relatively linear. This phenomenon can be seen in Table 5 – accuracy greatly improves as more sub-occultations are added, up to N = 5 sub-occultations, after which point performance remains relatively consistent."*

---

## Author Comment (AC2)

Thank you so much for reviewing our paper! We've copied your review here in blue. We've responded to each part of your review separately, in black. We've also printed our planned revisions in response to your comments here, in black and in italics.

Review 2:

GENERAL COMMENTS:

This paper presents a claimed novel technique for finding collocations between measurements from RO and passive nadir sounders. As the introduction highlights, these types of collocations have proven useful for various applications in the weather/atmospheric science community over the past years, thus this type of work presented is important and valuable to the community. The paper is well structured, clearly written, and has well placed/formatted figures. Results in the paper support their conclusion. I recommend it be accepted with minor revisions. Some of the specific comments below are just suggestions the authors can consider.

SPECIFIC COMMENTS:

Introduction – Is there any other publicly known/available code out there that does these sort of collocations – e.g. between different satellite tracks as referenced in your conclusions? This could be noted in the Introduction.

Yes, a few publicly available tools for finding satellite collocations already exist. As far as we are aware, all existing tools use brute-force methods, and some tools execute brute-force methods in parallel in the cloud in order to speed up collocation-finding.

We have added a line to the introduction noting this.

Original lines 44-45: *"We refer to collocation approaches similar to this as a brute force method."*

Revised lines 44-47: *"We refer to collocation approaches similar to this as a brute force method. Publicly available tools for collocating satellite data generally use brute-force approaches which are not specific  to the geometry of collocating GNSS RO and nadir-scanner soundings, and instead use parallelization and cloud computing to speed up collocation-finding (Chung et al., 2022; Smith et al., 2022; Wang et al., 2022)."*

Line 33 – Sentence starting "Intercomparison of RO …". It's not exactly correct to say "for the sake of validating the calibration of the infrared sounders…". It would be more exact to say "for the sake of validating the retrieved temperature products of the infrared sounders". The uncertainties involved with the radiative transfer model used to go between radiance and physical temperature doesn't (yet, from what I've seen) allow the RO to assess the calibration of the IR sounding instruments. If you have a reference for this it could certainly be included.

We appreciate the feedback, but would like to consider that studies exist that derive calibration offsets for microwave sounders. For example, the paper "Use of Radio Occultation to Evaluate Atmospheric Temperature Data from Spaceborne Infrared Sensors" (Yunck et al., 2009) compares AIRS atmospheric profiles to GPSRO profiles from CHAMP, SAC-C, and COSMIC, in order to derive bias offsets for AIRS. We now include a reference to this paper.

Original lines 33-35: *"Inter-comparison of RO and spectral thermal infrared sounders for the sake of validating the calibration of the infrared sounders has also been investigated (Feltz et al., 2017)"*

Revised lines 33-35: *"Inter-comparison of RO and spectral thermal infrared sounders for the sake of validating the calibration of the infrared sounders has also been investigated (Feltz et al., 2017; Yunck et al., 2009)"*

Line 75 (Intro of Section 2/2.1) – delta t and delta d should be more clearly defined, i.e. what time is used to define the "time" of the RO measurement (begin or end time)? What lat/long is defined as the location of the RO profile (perigee point)?

Good point -- we will clarify this. The "time" we use for each RO measurement is the start time, and the lat/long is the projection of the perigee (tangent point) onto Earth's surface.

Original lines 72-73: *"Collocations are defined as RO soundings that are separated from a passive nadir sounding by at most $\Delta t$ in time and $\Delta d$ in distance. First,..."*

Revised lines 80-83: *"Collocations are defined as RO soundings that are separated from a passive nadir sounding by at most $\Delta t$ in time and $\Delta d$ in distance. We consider the time corresponding to each RO sounding to be the start time of the RO measurement, and consider the position corresponding to each RO sounding to be the ray perigee (tangent) point projected onto Earth's surface. First,..."*

Table 3 – is a great way to show your results. Very organized and makes it easy to compare results from your collocation methods. You could consider adding the time match criterion in your table caption.

Yes, we've added the spatial and temporal criteria for collocation to the caption, for clarity.

Original line 351 (table 3 caption): *"Number of collocations by day, by satellite combinations."*

Revised line 384 (table 3 caption): *"Number of collocations by day, using $\Delta t = 600$ seconds as the temporal criterion and $\Delta d = 150$ km as the spatial criterion for collocation, by satellite combinations."*

Section 4.5/Table 4 – what time tolerance was tested to get the numbers for this Table? In hindsight I see it's the same as previous section, but maybe state again for explicitness.

Yes, we can state this explicitly.

Original line 358 (table 4 caption): *"Core-minutes required for computation by day, by satellite combinations (excluding data-loading)."*

Revised line 391 (table 4 caption): *"Core-minutes required for computation by day, using Δt = 600 seconds as the temporal criterion and Δd = 150 km as the spatial criterion for collocation, by satellite combinations (excluding data-loading)."*

Section 5 – You might consider making a comment about the geographic distribution of the collocations missed by the rotation method – your maps in previous sections illustrate this nicely for given days. However, adding a statement about the random geographic distribution of occultations (if true, which it looks like it is?) could (for some users) significantly strengthen the argument to use the rotation method.

Excellent point. This comment actually helped us correct a bad approximation in the rotation-collocation method! We had previously been using a constant satellite altitude when calculating scan distance, but the orbits of all the MW scanners considered in this paper are slightly eccentric, and so MW scanner altitude was slightly lower over the North Pole and slightly higher over the South Pole than what we had been modeling. As a result, we had a lot of false positives near the North Pole and a lot of false negatives over the South Pole.

We have now updated the rotation-collocation method to use the time-dependent altitude of the microwave satellite when calculating scan distance, and have updated our results. Now, the geographic distribution of misclassifications (all false positives + false negatives) roughly matches the geographic distribution of all collocations, as expected. The only deviation is that the mean latitude of false negatives is around 10 degrees south of the Equator, but the sample size is small, so it's difficult to draw conclusions about these distributions.

We added a figure showing a map of false positives and false negatives, as well as histograms over latitude and longitude for false positives, false negatives, and all collocations. We also added some comments about the distributions in sections 4.3 and 5.

New figure (on next page):

[Figure]

**Figure 6.** (a) Map of incorrect and missed predictions for all satellite combinations, (b) Histogram of latitude of all collocations, incorrect predictions, and missed predictions for all satellite combinations, (c) Histogram of longitude of all collocations, incorrect predictions, and missed predictions for all satellite combinations.

Addition to section 4.3 (lines 367-375): "*Figure 6(a) shows the geographic distribution of incorrect predictions and missed predictions. Figures 6(b) and 6(c) display the distribution of latitude and longitude, respectively, for incorrect predictions, missed predictions, and all collocations. The set of all collocations is roughly centered at the equator and prime meridian, with a mean latitude of 0.49∘, mean longitude of −1.69∘, standard deviation of latitude of 42.2∘, and standard deviation of longitude of 104.1∘. The distribution of incorrect predictions is similar, with a mean latitude of 2.82∘, mean longitude of 4.52∘, standard deviation of latitude of 42.2∘, and standard deviation of longitude of 103.4∘. The distribution of missed predictions, however, is centered slightly south of the equator; it has a mean latitude of −12.58∘, mean longitude of −10.5∘, standard deviation of latitude of 34.1∘, and standard deviation of longitude of 105.4∘. The sample size (n = 116) of missed predictions is small, however, which makes it difficult to evaluate the significance of this small shift in geographic distribution.*"

Addition to section 5 (lines 467-468): "*Currently, the geographic distribution of the soundings misclassified by the rotation-collocation algorithm roughly matches the geographic distribution of collocated soundings, as shown in Figure 6.*"

Addition to section 5.1 (lines 492-3): "The authors also plan to further investigate the geographic distribution of collocations missed by the rotation-collocation method."

TECHNICAL COMMENTS/CORRECTIONS:

Line 68 – "define" should be "defines"

Good catch.

Original lines 67-68: *"Section 3 describes the data sets that will be used in the study and define the experimental setup."*

Revised lines 75-76: *"Section 3 describes the data sets that will be used in the study and defines the experimental setup."*

Line 95 – should Section 2.1 be 2.1.1?

Yes, it should.

Original line 95: *"This approach is similar to that of the brute-force method discussed in §2.1 but with narrowed windowing in time."*

Revised line 104: *"This approach is similar to that of the brute-force method discussed in §2.1.1 but with narrowed windowing in time."*

Line 146 – define ECI acronym

Yes -- we should define the ECI acronym before using it.

Original lines 143-144: *"...the coordinates $x_{ECI}(t)$, $y_{ECI}(t)$, $z_{ECI}(t)$ are Cartesian coordinates of a location in an Earth-centered inertial coordinate system."*

Revised lines 152-153: *"...the coordinates $x_{ECI}(t)$, $y_{ECI}(t)$, $z_{ECI}(t)$ are Cartesian coordinates of a location in an Earth-centered inertial (ECI) coordinate system"*

Line 252 – "four collocation-finding methods" – only 3 lines shown in Fig 2(a)

Both brute-force methods find the same collocations, and are represented by the same line in Figure 2(a). We will make this explicit.

Original lines 251-254: *"In Figure 2(a), we show the collocations between COSMIC-2 and NOAA-20 by day found by each of our four collocation-finding methods. The rotation-collocation algorithm with sub-occultations (orange) and the linearized rotation-collocation algorithm (light green) find slightly more collocations on each day than the brute-force algorithm (blue)…"*

Revised lines 267-271: *"In Figure 2(a), we show the collocations between COSMIC-2 and NOAA-20 by day found by each of our four collocation-finding methods. Both brute-force methods yield identical results, and so both methods are represented in Figure 2(a) by the same blue line. The rotation-collocation algorithm with sub-occultations (orange) and the linearized rotation-collocation algorithm (light green) find slightly more collocations on each day than the brute-force algorithms (blue)…"*

Line 347 – "non" to "none"?

Yes, "non" was a typo.

Original lines 346-347: *"There are no collocations between Metop-B-AMSU and Metop-C-GRAS, and non between Metop-C-AMSU and Metop-B-GRAS."*

Revised lines 380-381: *"There are no collocations between Metop-B-AMSU and Metop-C-GRAS, and none between Metop-C-AMSU and Metop-B-GRAS."*

Line 433 – get rid of "that"?

Yes.

Original lines 433-434: *"Cohosted instruments on Metop-B and Metop-C greatly increase the percentage of soundings that are collocated, and that cohosting MW and RO instruments is a powerful tool for increasing the number of collocations."*

Revised lines 477-478: *"Cohosted instruments on Metop-B and Metop-C greatly increase the percentage of soundings that are collocated, and cohosting MW and RO instruments is a powerful tool for increasing the number of collocations."*